# EGFR signaling promotes self-renewal through the establishment of cell polarity in *Drosophila* follicle stem cells

Angela Castanieto[1,2], Michael J Johnston[1,2], Todd G Nystul[1,2]*

[1]Department of Anatomy, University of California, San Francisco, San Francisco, United States; [2]Center for Reproductive Sciences, Department of Obstetrics, Gynecology and Reproductive Sciences, University of California, San Francisco, San Francisco, United States

**Abstract** Epithelial stem cells divide asymmetrically, such that one daughter replenishes the stem cell pool and the other differentiates. We found that, in the epithelial follicle stem cell (FSC) lineage of the *Drosophila* ovary, epidermal growth factor receptor (EGFR) signaling functions specifically in the FSCs to promote the unique partially polarized state of the FSC, establish apical–basal polarity throughout the lineage, and promote FSC maintenance in the niche. In addition, we identified a novel connection between EGFR signaling and the cell-polarity regulator liver kinase B1 (LKB1), which indicates that EGFR signals through both the Ras–Raf–MEK–Erk pathway and through the LKB1–AMPK pathway to suppress apical identity. The development of apical–basal polarity is the earliest visible difference between FSCs and their daughters, and our findings demonstrate that the EGFR-mediated regulation of apical–basal polarity is essential for the segregation of stem cell and daughter cell fates.

*For correspondence: todd. nystul@ucsf.edu

**Competing interests:** The authors declare that no competing interests exist.

## Introduction

Adult stem cell divisions produce asymmetric outcomes such that one daughter self-renews while the other goes on to differentiate. Although the signals that specify these different cell fates act directly on stem cells and their immediate daughters, the process of differentiation in the non-stem cell daughters can be gradual, sometimes occurring over the course of several cell divisions. Indeed, in many stem cell lineages, the non-stem cell daughter initially has the potential to re-enter the niche and become a stem cell, indicating that it does not immediately commit to the fully differentiated fate (*Morrison and Spradling, 2008*; *Simons and Clevers, 2011*). Instead, newly produced stem cell daughters are likely in a state of flux, undergoing a wide range of molecular and cytological changes. Previous studies have exploited visible differences in the cellular and sub-cellular morphology of stem cells and their daughters to investigate the signals that govern the segregation of cell fates (*Xie and Spradling, 1998*; *Lim et al., 2000*; *Kiger et al., 2001*; *Ohlstein and Spradling, 2007*; *Yin et al., 2013*). In several different *Drosophila* and mammalian epithelia, the stem cells have a constricted or immature apical domain relative to the differentiated cell types in the tissue (*Ohlstein and Spradling, 2006*; *Buske et al., 2011*; *Huo and Macara, 2014*; *Kronen et al., 2014*). Therefore, signals that regulate the development of cell polarity may play an important role in the segregation of epithelial stem cell and daughter cell fates. However, the relationship between cell polarity and differentiation in epithelial stem cell lineages is poorly understood.

In this study, we investigated the signals that promote stem cell maintenance in the niche and apical–basal cell polarity in the epithelial follicle stem cells (FSCs) of the *Drosophila* ovary. Two FSCs are maintained within a structure at the tip of each ovariole, called the germarium (*Figure 1A*)

**eLife digest** A stem cell is a special cell that divides to produce another stem cell, plus a cell that goes on to perform a specific role in the body. The process by which this second cell becomes a specific type of cell is called differentiation. The body contains many different types of stem cells, such as neural stem cells, which go on to form the nervous system, and epithelial stem cells, which give rise to various types of surfaces in the body, such as the skin and the lining of the intestine.

Many types of epithelial cells are polarized, which means they have three distinct sides or domains: a basal domain that faces the underlying tissue; an apical domain on the opposite side; and a lateral domain on the side in between the apical and basal domains. The details of how cell polarity is established in epithelial cells are not fully understood, but it is thought to have its origins in the division of epithelial stem cells.

Now, by studying follicle stem cells in the ovaries of fruit flies, Castanieto et al. have shown that a process called EGFR signaling (which is short for epidermal growth factor receptor signaling) has a central role in establishing the difference between the stem cell and the cell that differentiates. EGFR signaling does this, in part, by promoting a 'partially polarized state' in the stem cells: this state is characterized by the presence of a basal domain and a lateral domain but no apical domain.

In fully polarized cells, the apical and lateral domains work together to ensure that all three domains remain separated on the surface of the cell, so it was surprising to find that the stem cell could maintain basal and lateral domains without an apical domain. Castanieto et al. propose that this feat is achieved by EGFR signaling, which activates a multiple number of proteins, including one called LKB1 that is known to regulate cell polarity.

This work strongly suggests that that changes in cell polarity are among the earliest differences to arise between epithelial stem cells and differentiating cells. In the future, it will be important to determine whether these differences in cell polarity cause the stem cells and the differentiating cells to take on different roles in the tissue. For example, it may be that the lack of an apical domain in the stem cells shields them from signals in the tissue that promote differentiation, thus allowing them to remain undifferentiated. Conversely, the development of an apical domain in the differentiating cells may expose them to signals that promote their differentiation, and also allow them to form a barrier and perform the other roles of epithelial tissue.

(*Margolis and Spradling, 1995*). A population of stromal escort cells located just anterior to the FSCs forms the niche, providing essential self-renewal ligands to the FSCs (*Song and Xie, 2003*; *Sahai-Hernandez and Nystul, 2013*), and also supports early germ cell cyst development (*Kirilly et al., 2011*; *Eliazer et al., 2014*). As germ cell cysts mature, they move out of the escort cell region into the follicle epithelium, and each FSC divides approximately once per incoming cyst (*Nystul and Spradling, 2010*). Newly produced FSC daughter cells, called prefollicle cells, move away from the niche either toward the posterior or across the germarium toward the opposite FSC niche before incorporating into the follicle epithelium (*Nystul and Spradling, 2007*). This well-characterized tissue architecture makes it possible to readily identify and genetically manipulate FSCs and their immediate daughter cells within intact ovarioles.

The development of cell polarity in the follicle epithelium is a multi-step process (*Franz and Riechmann, 2010*; *Kronen et al., 2014*). The FSCs have a lateral domain (which includes Discs large (Dlg), Lethal giant larvae (Lgl), and Scribble), a basal domain (defined by the localization of integrins to the surface that contacts the basement membrane), but no discrete apical domain. Instead, apical determinants are either undetectable or diffusely localized throughout the FSCs. In contrast, nascent apical domains are visible in adjacent, differentiating prefollicle cells. The apical–lateral determinant, Bazooka (Baz, the *Drosophila* homolog of Par-3), is the first marker to become visible on the cell membrane, followed by atypical protein kinase C (aPKC), which localizes to the apical surface and positions Baz at the apical–lateral junctions. Further downstream, proteins in the Crumbs complex colocalize with aPKC and reinforce the apical identity. Adherens junctions also relocalize during follicle cell differentiation from a broad band along the anterior lateral surface of FSCs to discrete puncta at the apical–lateral junctions in differentiated follicle cells.

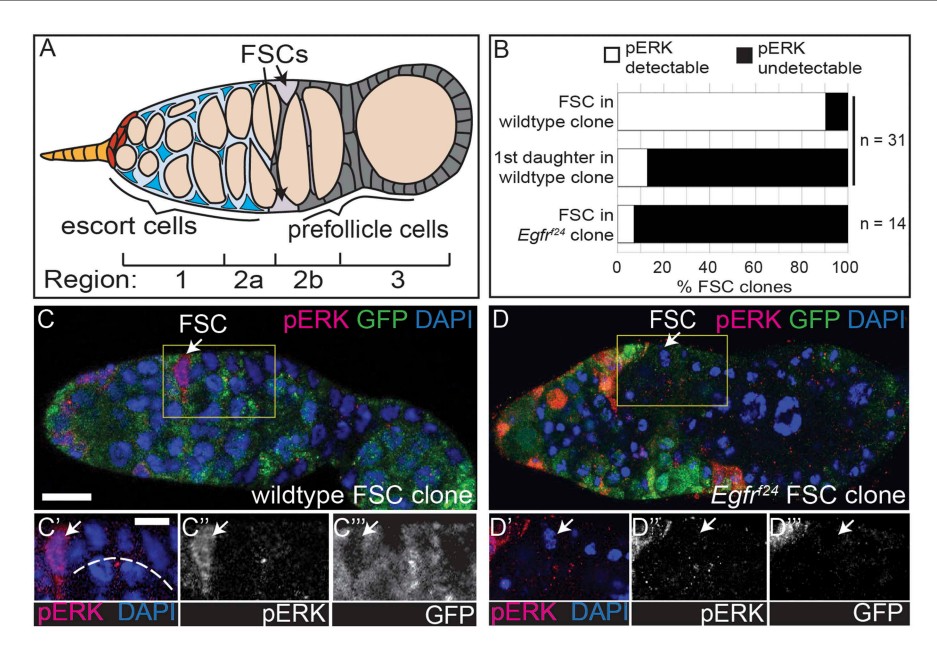

**Figure 1**. The EGFR pathway is upregulated specifically in FSCs. (**A**) Diagram of the germarium of the Drosophila ovary. The germarium is divided into four regions as indicated; anterior is to the left. Two follicle stem cells (FSCs, light grey) are maintained in the germarium at the Region 2a/2b border. Escort cells (blue) are anterior to the FSCs and support development of the early germline (orange). As they mature, germline cysts move posteriorly out of Region 2a and into the follicle epithelium. Each FSC divides once per incoming cyst, producing prefollicle cells (dark grey) that encapsulate the germline as it moves into Region 2b. (**B**) Quantification of pErk staining of FSCs and prefollicle cells just downstream from the niche within a wildtype or *Egfr^f24* FSC clone. (**C–D**) Wildtype (**C**) or *Egfr^f24* (**D**) FSC clones stained for pErk (red), GFP (green) and DAPI (blue). Boxed regions of **C–D** are magnified in **C'–C'''** and **D'–D'''**. White arrows indicate the FSC, which is the anterior-most GFP^(–) follicle cell in the clone. White dashed line in (**C'**) indicates prefollicle cells in which pErk is undetectable compared to the FSC. Scale bar represents 5 μm in **C–D** and 1 μm in magnified insets.

The following figure supplement is available for figure 1:

**Figure supplement 1**. pErk staining in the ovariole **A**.

Once cell polarity is established in epithelial tissues, it is maintained through a highly conserved, self-sustaining process of mutual repression between the apical and lateral protein complexes (*Betschinger et al., 2003*; *Bilder et al., 2003*; *Tanentzapf and Tepass, 2003*). However, since these complexes are absent or immature during the establishment of cell polarity, additional signals are required at earlier stages. One such signal is provided by liver kinase B1 (LKB1), which regulates multiple proteins involved in the establishment of cell polarity, including AMP-activated protein kinase (AMPK) (*Nakano and Takashima, 2012*). LKB1 is activated by protein kinase A (PKA) and Par-1, and is required for follicle cell polarity (*Martin and St Johnston, 2003*; *Haack et al., 2013*). Epidermal growth factor receptor (EGFR) signaling may also be important for follicle cell polarity because a global reduction of the function of *Egfr* or of the downstream EGFR pathway modifiers *brainiac* or *egghead* disrupts the architecture of the follicle epithelium; however cell-polarity markers were not investigated (*Goode et al., 1992*, *1996*). Here we show that EGFR signaling promotes FSC maintenance in the niche, that EGFR is required specifically in FSCs to establish cell polarity throughout the FSC lineage, and that EGFR signals through both the canonical Ras–Raf–MEK–Erk pathway and through LKB1 and AMPK to suppress apical identity.

## Results

### The EGFR pathway is upregulated specifically in FSCs

To determine which cells within the early FSC lineage have active EGFR signaling, we stained for the dual-phosphorylated extracellular signal-related kinase (pErk), one of the downstream effectors of the

canonical EGFR pathway. Consistent with published studies, we found that pErk was detectable in escort cells (*Liu et al., 2010*), sporadic follicle cells in the germarium, and most follicle cells surrounding mid-stage follicles (*Van Buskirk and Schupbach, 1999*; *Chen et al., 2013*) (*Figure 1—figure supplement 1A*). In addition, we noticed bright pErk staining in cells at the position of the FSC. As FSCs can be unambiguously identified as the anterior-most cell in an FSC clone induced in adult flies by mitotic recombination, we generated FSC clones marked by the absence of green florescent protein (GFP) and stained for pErk. Indeed, we found that bright pErk staining was detected in 90% (n = 28/31) of FSCs identified within an FSC clone, but was completely undetectable in 87% (n = 27/31) of prefollicle cells just downstream of the niche (*Figure 1B,C*). In the remaining 13% of prefollicle cells just downstream of the niche (n = 4/31), pErk was also detectable, which could be due to the perdurance of the pErk signal in cells that had recently exited the niche (*Figure 1—figure supplement 1B*). To determine whether this pErk signal is dependent upon EGFR, we generated FSC clones that are homozygous for *Egfr^{f24}*, a loss-of-function allele, and stained for pErk. Indeed, we found that pErk was undetectable in the FSC in 93% (n = 13/14) of *Egfr^{f24}* FSC clones (*Figure 1B,D*). Taken together, these results indicate that the EGFR pathway is active in FSCs and downregulated in prefollicle cells that have moved downstream from the FSC niche.

## EGFR is required for FSC maintenance in the niche

Given the specificity of EGFR signaling in the FSC, we hypothesized that EGFR is required for FSC maintenance in the niche. To test this hypothesis, we performed a standard assay (*Song and Xie, 2002*) in which clones are generated in adult ovaries, and the frequencies of ovarioles with 0, 1, or 2 clonally marked FSCs are quantified at multiple time points after clone induction. In this assay, ovarioles that start out as mosaic (1 marked FSC) become fully marked (2 marked FSCs) or fully unmarked (0 marked FSCs) when the daughter of one FSC replaces the other FSC. Thus, FSC turnover causes a decrease in the frequency of mosaic ovarioles, and an increase in the frequencies of fully marked and fully unmarked ovarioles. If both the marked and unmarked FSCs are wildtype, they will replace each other at equal rates so the frequencies of fully marked and fully unmarked ovarioles increase at approximately equal rates. If the clonally marked FSCs are mutant for a gene that is required for FSC maintenance in the niche, the marked FSCs will be rapidly lost, causing a disproportionate increase in the frequency of fully unmarked ovarioles. Conversely, if the clonally marked FSCs contain a genetic modification that enhances their ability to occupy the niche or replace wildtype FSCs, unmarked FSCs will be preferentially lost, causing a disproportionate increase in the frequency of fully marked ovarioles.

We generated either wildtype (control group) or *Egfr^{f24}* (experimental group) GFP^{(−)} clones (*Figure 2A–B*) under identical conditions (See 'Materials and methods') and assayed for FSC clone frequencies at 4, 7, and 11 days post clone induction (dpci). It is more common to assay for FSC clone frequencies at 7, 14, and 21 dpci, but because *Egfr^{f24}* FSC clones were extremely rare at late time points we chose this earlier set of time points. Nonetheless, we found significantly fewer (p < 0.02) mosaic ovarioles in the experimental group compared to the control group at all three early time points (*Figure 2C* and *Figure 2—figure supplement 1*). Moreover, while the frequency of fully marked ovarioles in the control group increased from 0% at 4 dpci (n = 0/157) to 10% at 11 dpci (n = 9/87), no fully marked ovarioles were observed in the experimental group at any time point (*Figure 2C*). Thus, *Egfr* is required for FSC maintenance in the niche.

To verify that the low frequency of *Egfr^{f24}* FSC clones was not due to a lower rate of clone induction, we also quantified the FSC clone frequencies in the control and experimental groups at 2 dpci, which is the earliest time at which GFP^{(−)} FSC clones can be detected. We found that even at 2 dpci the frequency of *Egfr^{f24}* FSC clones in the experimental group was still significantly lower (p < 0.01) than that of the control group (*Figure 2C*). Therefore, we next measured the frequency of all FSC and follicle cell clones in the germarium of each ovariole at 2 dpci and found that the combined frequencies of FSC clones and prefollicle cell clones (i.e. those originating in follicle cells downstream of the FSC) were nearly the same in the control and experimental groups (*Figure 2D*). This suggests that the low frequency of *Egfr^{f24}* FSC clones at 2 dpci was due to a very rapid loss of *Egfr^{f24}* FSCs and not a lower rate of clone induction, as explored further in the next section.

Lastly, since loss of *Egfr* inhibits FSC maintenance in the niche, we tested whether constitutively active EGFR signaling had the opposite effect. To test this, we used the MARCM system (*Lee and Luo, 2001*), which makes it possible to express a transgene of interest in all of the cells in the

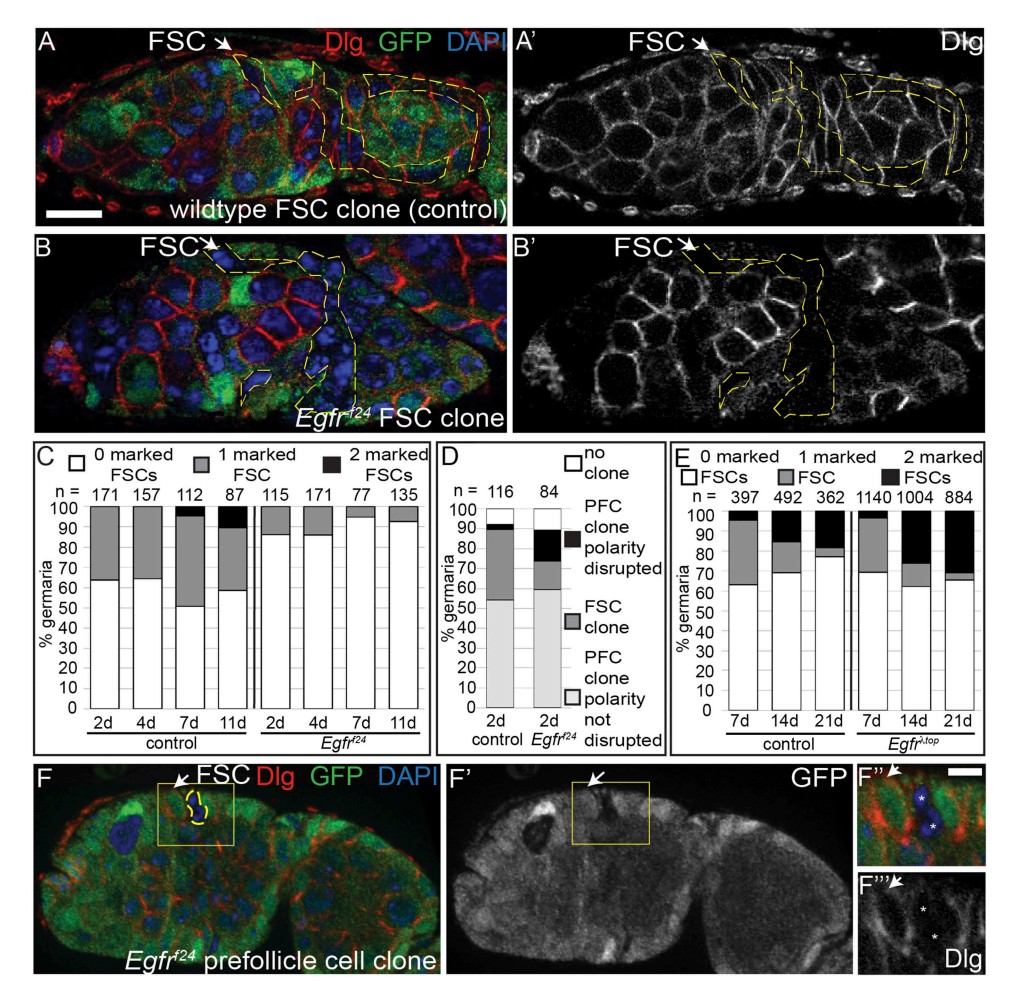

**Figure 2**. EGFR is required for FSC maintenance in the niche. (**A–B**) Germaria with a mature wildtype (**A**) or *Egfr*[f24] (**B**) GFP[(−)] FSC clone stained for Dlg (red) and GFP (clone marker, green). (**C–E**) Graphs indicating the frequencies of the *Egfr*[f24] or control FSC clones at 2, 4, 7, and 11 dpci (**C**); all *Egfr*[f24] or control clones, including polarity-defective *Egfr*[f24] prefollicle cell (PFC) clones, at 2 dpci (**D**); and the *Egfr*[λtop] or control FSC clones at 7, 14, and 21 dpci (**E**). (**F**) Polarity-defective *Egfr*[f24] prefollicle cell clone at 2 dpci, stained for Dlg (red) and GFP (green); **F′** shows the GFP channel alone; boxed regions are magnified in **F″–F‴**. GFP[(−)] clones are indicated by dashed yellow lines, and by white asterisks in **F″–F‴**. White arrows indicate the position of the FSC niche. All tissues stained with DAPI (blue). Anterior is to the left. Scale bar represents 5 µm in **A–F** and 1 µm in magnified insets.

The following figure supplements are available for figure 2:

**Figure supplement 1**. Quantification of marked control and *Egfr*[f24] FSC clone frequencies at 2, 4, 7, and 11 dpci.

**Figure supplement 2**. Quantification of marked control and *Egfr*[λtop] FSC clone frequencies at 7, 14, and 21 dpci.

clone. We generated groups of flies in which the clonally marked cells were either wildtype (control group) or overexpressing a constitutively activated allele of *Egfr* (*Egfr*[λtop]) (experimental group) (*Queenan et al., 1997*). We induced clones in paired control and experimental groups, and assayed for FSC clone frequencies at 7, 14, and 21 dpci. Whereas the frequency and distribution of FSC clones was similar in both groups at 7 dpci (p > 0.33), the frequency of fully marked ovarioles was significantly higher in the experimental group compared to the control group at 21 dpci (18% in wildtype vs 31% in *Egfr*[λtop], p < 0.02) (*Figure 2E* and *Figure 2—figure supplement 2*). This indicates that FSCs and prefollicle cells with constitutively active EGFR signaling are hypercompetitive for the niche relative to the wildtype FSCs in the same tissue.

## EGFR is required specifically in the FSC for establishment of epithelial polarity

Interestingly, all *Egfr^f24* FSC clones and a subset of early *Egfr^f24* prefollicle cell clones had severe morphological defects that suggested a loss of cell polarity. Indeed, the *Egfr^f24* cells in these clones failed to incorporate into the follicle epithelium and did not encapsulate germ cell cysts (*Figure 2B*). To determine whether these cells had polarity defects, we stained ovarioles with *Egfr^f24* FSC clones for markers of apical, lateral, and basal identity. We found that the lateral Dlg was undetectable on the cell membranes in 100% (n = 38/38) of *Egfr^f24* FSC clones (*Figure 3F* and *Table 1*). In addition, we found that the apical marker aPKC, the apical–lateral marker Baz, and the adherens junction component DE-cadherin (DE-cad) were also undetectable in all *Egfr^f24* FSC clones (*Figure 3G–I*). Moreover, the basal marker β-integrin was detectable in the cytoplasm but not on the cell membrane in all *Egfr^f24* FSC

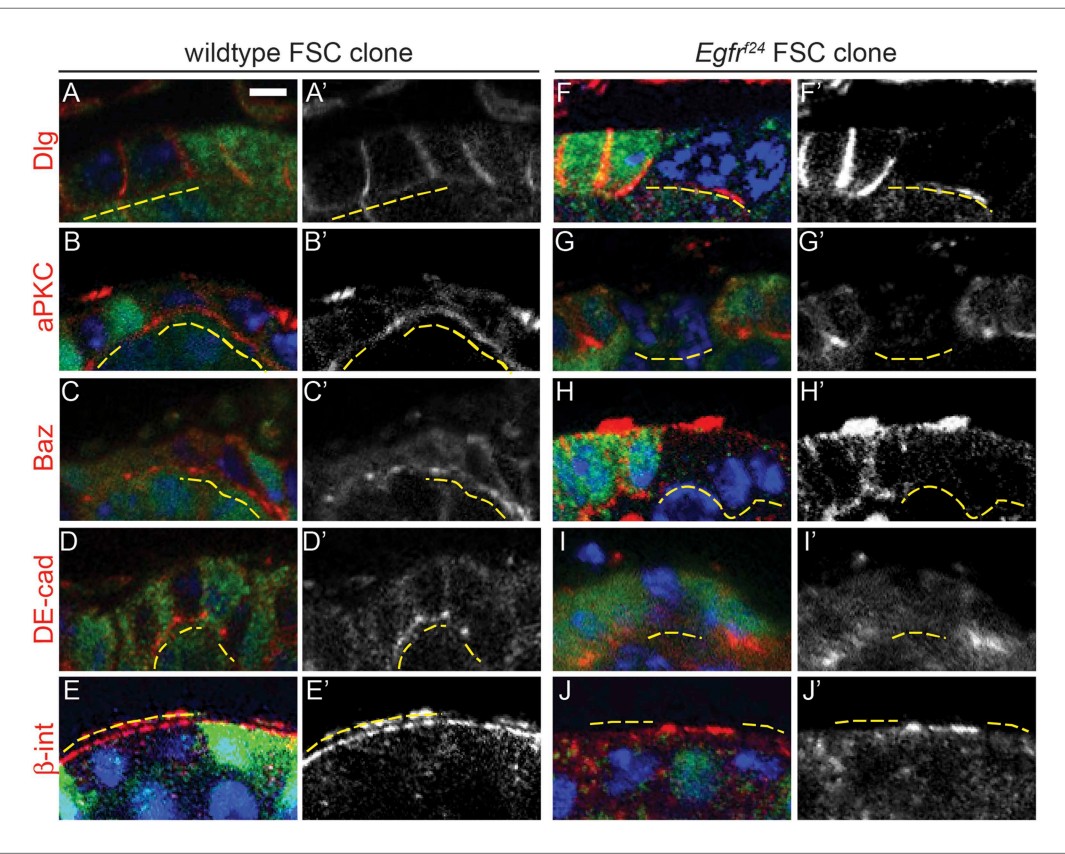

**Figure 3**. *Egfr^f24* FSC clones have epithelial polarity defects. (**A–J**) Wildtype (**A–E**) and *Egfr^f24* (**F–J**) FSC clones stained for polarity markers (red) Dlg (**A** and **F**), aPKC (**B** and **G**), Baz (**C** and **H**), DE-cad (**D** and **I**) and β-integrin (β-int) (**E** and **J**); GFP (green); and DAPI (blue). Panels **A'–J'** show the red channel only. All polarity markers are able to localize properly in wildtype GFP^(–) FSC clones and in the GFP^(+) follicle cells within germaria that contain either wildtype or *Egfr^f24* FSC clones; all polarity markers are undetectable in GFP^(–) *Egfr^f24* FSC clones. GFP^(–) clones are indicated by dashed yellow lines. Images are oriented with the apical surface of the follicle cells on the bottom. Scale bar represents 1 µm.

The following figure supplements are available for figure 3:

**Figure supplement 1**. Positively marked *Egfr^f2* FSC clones have epithelial polarity defects.

**Figure supplement 2**. Quantification of the frequency of polarity phenotypes in positively marked control FSC clones, Egfrf2 FSC clones, and Egfrf2 prefollicle cell clones.

**Figure supplement 3**. Polarity defects in *Egfr^DN* follicle cells.

**Table 1.** Quantification of the frequency of polarity phenotypes in control FSC clones, *Egfr^f24* FSC clones, and *Egfr^f24* prefollicle cell clones

|  | polarity not disrupted | polarity disrupted |
|---|---|---|
| Wildtype FSC clone | 96%, 103/107 | 4%, 4/107 |
| *Egfr^f24* prefollicle cell clone | 94%, 82/87 | 6%, 5/87 |
| *Egfr^f24* FSC clone | 0%, 0/38 | 100%, 38/38 |

Values reflect both the percent and fraction of each clone type in which polarity is disrupted or not disrupted as indicated.

clones (*Figure 3J*). In contrast, polarity was not disrupted in 96% (n = 103/107) of negatively marked wildtype FSC clones (*Figure 3A–E* and *Table 1*), and we consistently found that all of these cell-polarity markers were properly localized on the cell membranes of *Egfr^+/f24* follicle cells in germaria with *Egfr^f24* clones (*Figure 3F–J*). In addition, using the MARCM system we generated positively marked FSC clones homozygous for another loss of function allele, *Egfr^f2*, and found that 94% (n = 17/18) had an identical phenotype as negatively marked *Egfr^f24* FSC clones, while polarity was not disrupted in 99% (n = 83/84) of positively marked wildtype FSC clones (*Figure 3—figure supplement 1C–F* and *Figure 3—figure supplement 2*). Lastly, we found that expression of a dominant-negative allele of *Egfr* (*Egfr^DN*) using a follicle cell-specific driver 109-30-Gal4 (Figure 3—figure supplement 3A) (*Hartman et al., 2010*) phenocopied the polarity defects we observed in *Egfr^f24* clones, albeit with a lower penetrance (46% of germaria, n = 118/255, Figure 3—figure supplement 3B–C). Together, these controls verify that the phenotypes we observed are not due to the genetic background, that they are not an artifact of the dissection process (*Haack et al., 2013*), and that the loss of cell polarity in *Egfr^f24* and *Egfr^f2* FSC clones is cell autonomous.

To determine whether *Egfr^f24* FSC clones retained other markers of epithelial identity we stained for FasIII, which is commonly used to identify follicle cells, and found that it was consistently detectable on the cell membrane (*Figure 4A*). Likewise, we found that Traffic jam, a transcription factor that is specific for somatic cells that contact the germline (*Li et al., 2003*), was also unaffected (*Figure 4A*). To determine whether the observed polarity defects were associated with apoptosis, we stained ovarioles with *Egfr^f24* FSC clones and ovarioles from *Egfr^f24/+* siblings as a control for cleaved Caspase 3 (Cas3). We found that while all *Egfr^f24* FSC clones had polarity defects, the frequency of *Egfr^f24* FSC clones with Cas3+ follicle cells in the germarium (17%, n = 3/16) was comparable to the frequency observed in the sibling controls (15%, n = 6/41, *Figure 4C*). Cas3 was frequently detectable in the polar cells of newly budded follicles, as expected (*Khammari et al., 2011*) (*Figure 4B*). Thus, the polarity defect in *Egfr^f24* FSC clones is not likely to be due to a loss of follicle cell identity or the induction of apoptosis.

Although nearly all *Egfr^f24* and *Egfr^f2* FSC clones had polarity defects, Dlg was not disrupted in 94% (n = 82/87) and 95% (n = 121/127) of *Egfr^f24* and *Egfr^f2* prefollicle cell clones, respectively (*Table 1*,

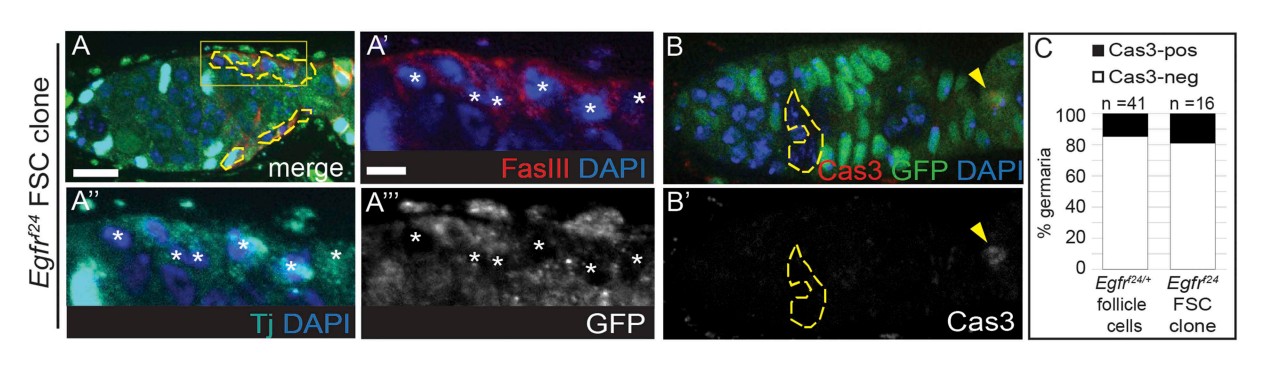

**Figure 4**. Loss of EGFR does not cause cell death or loss of follicle cell identity. (**A**) *Egfr^f24* FSC clone with normal FasIII (red) and Traffic jam (Tj) (cyan) in the clone, indicated by white asterisks in the magnified regions in (**A'**–**A'''**). (**B**) *Egfr^f24* FSC clone with a Cas3-positive cell (red, yellow arrowhead) in the polar region of a newly budded follicle, but not in the clone. Panel **B'** shows the red channel only. (**C**) Graph indicating the frequency of Cas3-positive follicle cells in *Egfr^f24/+* control germaria or in *Egfr^f24* FSC clones. GFP^(−) clones are indicated by dashed yellow lines. All tissues stained with DAPI (blue). Anterior is to the left. Scale bar represents 5 µm in (**A**–**B**), and 1 µm in magnified insets.

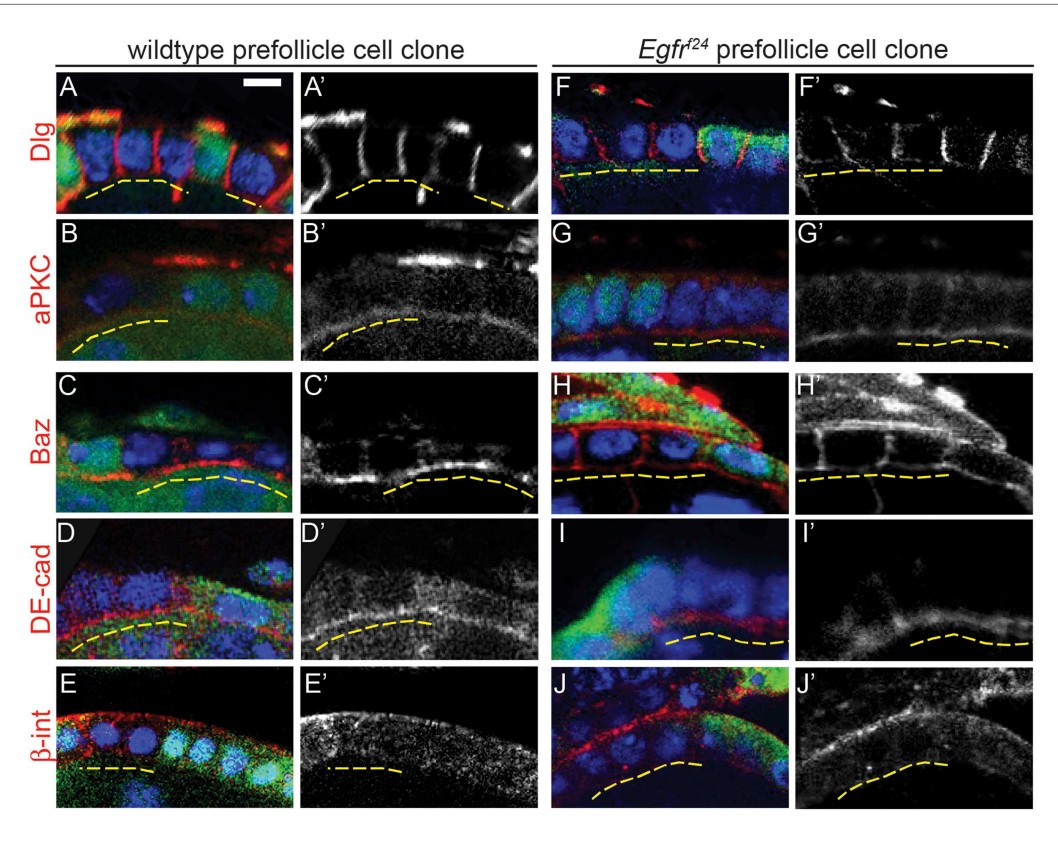

**Figure 5**. *Egfr^f24* prefollicle cell clones do not have epithelial polarity defects. (**A–J**) Wildtype (**A–E**) and *Egfr^f24* (**F–J**) prefollicle cell clones stained for polarity markers (red) Dlg (**A** and **F**), aPKC (**B** and **G**), Baz (**C** and **H**), DE-cad (**D** and **I**) and β-int (**E** and **J**); GFP (green); and DAPI (blue). Panels **A'–J'** show the red channel only. All polarity markers are properly localized in both wildtype and *Egfr^f24* GFP^(–) prefollicle cell clones. GFP^(–) clones are indicated by dashed yellow lines. Images are oriented with the apical surface of the follicle cells on the bottom. Scale bar represents 1 μm.

The following figure supplement is available for figure 5:

**Figure supplement 1**. pErk is absent from *Egfr^f24* prefollicle cell clones.

*Figure 5F*, *Figure 3—figure supplement 1A–B*, and *Figure 3—figure supplement 2*; see 'Materials and methods' for a description of FSC clones vs prefollicle cell clones). In addition, aPKC, Baz, DE-cad, and β-integrin were not substantially disrupted in large wildtype and *Egfr^f24* prefollicle cell clones (*Figure 5*). Lastly, we found that pErk was undetectable in nearly all *Egfr^f24* prefollicle cell clones (*Figure 5—figure supplement 1*), verifying that the lack of polarity defects in these clones was not due to a perdurance of EGFR signaling. Together these data indicate that EGFR is not needed for the continued establishment or maintenance of cell polarity in follicle cells downstream of the FSC.

At 2 dpci, 15% (n = 13/84, *Figure 2D*) of *Egfr^f24* clones in the germarium that did not include a cell in the FSC niche (*Figure 2F*) had polarity defects identical to those of *Egfr^f24* FSC clones. Since *Egfr^f24* FSCs are rapidly lost from the niche, it is likely that these polarity-defective prefollicle cell clones originated from a recently replaced FSC, but had not yet moved out of the germarium at this early time point. Indeed, the occurrence of early *Egfr^f24* prefollicle cell clones with disrupted polarity at 2 dpci almost fully accounts for the reduced number of *Egfr^f24* FSC clones compared to the wildtype control FSC clones (*Figure 2D*). These observations are consistent with a very rapid loss of *Egfr^f24* FSCs from the niche and demonstrate that the cell polarity defects arise within 2 dpci. Collectively, these data indicate that *Egfr* is required specifically in FSCs to establish cell polarity throughout the FSC lineage.

## Downregulation of EGFR activity is required for apical polarization of early follicle cells

To investigate whether the downregulation of EGFR signaling in prefollicle cells is necessary for the establishment of the apical domain, we expressed *Egfr*$^{\lambda top}$ throughout the early FSC lineage using 109-30-Gal4, which activated pErk throughout the early follicle cell lineage (*Figure 6—figure supplement 1A*), and stained for cell polarity markers. In wildtype germaria, apical domains begin to form in Region 2b of the germarium, just downstream from the FSC niche, but the follicle cells do not acquire an organized, cuboidal shape with a clear apical surface until Region 3 of the germarium (*Figure 6A* and *Figure 6—figure supplement 1B*). We found that Region 3 cysts were always present in germaria from control flies, but that 21% (n = 25/119) of the germaria with constitutively active EGFR signaling were elongated and completely lacked a Region 3 cyst (*Figure 6—figure supplement 1C*). In nearly all of the remaining germaria (70%, n = 83/119), Region 3 cysts were present but aPKC was delocalized

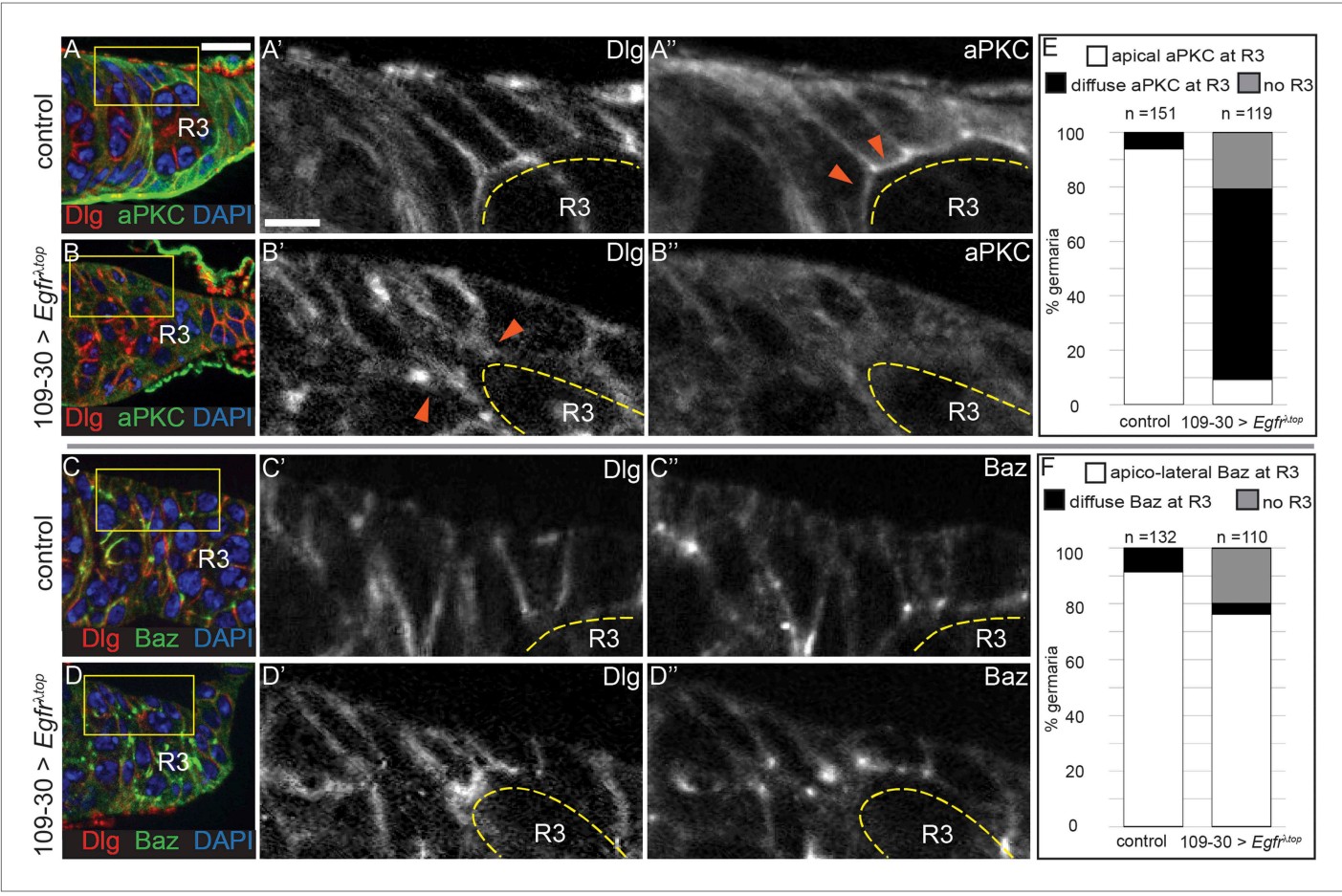

**Figure 6**. Constitutive activation of EGFR disrupts prefollicle cell apical polarity. (**A–D**) Control germaria containing UAS-*Egfr*$^{\lambda top}$ but no Gal4 driver (**A** and **C**) and experimental germaria in which *Egfr*$^{\lambda top}$ is expressed in follicle cells under the control of 109-30-Gal4 (**B** and **D**) and stained for Dlg (red), DAPI (blue), and either aPKC (green, **A–B**) or Baz (green, **C–D**). Follicle cells along the Region 3 cyst (R3, yellow dashed line) of control germaria have a cuboidal shape with a clear apical surface (**A″**, orange arrowheads); aPKC localizes to the apical surface (**A″**), Baz localizes to apical–lateral junctions (**C″**), and Dlg localizes to lateral surfaces (**A′** and **C′**). In germaria expressing *Egfr*$^{\lambda top}$ in which the R3 cyst is present, cells have a pointed shape and form narrow contacts with the germline (**B′**, orange arrowheads). In addition, aPKC is delocalized from the cell surface of follicle cells (**B″**), but Dlg is detectable on the cell membrane (**B′** and **D′**) and Baz localizes to apical–lateral junctions (**D″**). (**E–F**) Graphs indicating the frequencies of control or experimental germaria with no R3 cyst, or with localized or delocalized aPKC (**E**) or Baz (**F**) in follicle cells along the R3 cyst. Boxed regions of (**A–D**) are magnified in **A′–D″**. Anterior is to the left. Scale bar represents 5 μm in **A–D** an 1 μm in magnified insets.

The following figure supplement is available for figure 6:

**Figure supplement 1**. Expression of *Egfr*$^{\lambda top}$ in follicle cells.

from the cell surface of follicle cells surrounding these cysts (*Figure 6B,E* and *Figure 6—figure supplement 1D*). In addition, many of the follicle cells in this region were not cuboidal but instead had a more pointed shape that resembled FSCs or early prefollicle cells (*Figure 6B*). Interestingly, Dlg was still detectable on the cell membranes of follicle cells with constitutively active EGFR signaling (*Figure 6B,D*), indicating that lateral identity was largely unaffected in these cells. In addition, bright puncta of Baz staining were still visible on the cell membrane near sites of contact with the germline in 76% (n = 86/110) of these germaria, suggesting that apical–lateral identity is also largely unaffected (*Figure 6C–D,F*). Consistent with this finding, Baz remains localized to the apical–lateral junctions in aPKC$^{-/-}$ follicle cells (*Morais-de-Sá et al., 2010*). Therefore, downregulation of EGFR signaling is required for the formation of apical domains in prefollicle cells.

## Ras and LKB1 are downstream of EGFR and are required to establish epithelial polarity

To investigate the mechanism by which EGFR signaling promotes the establishment of cell polarity in follicle cells, we searched for other genes that were required in FSCs but not prefollicle cells for cell polarity. First, we tested *Ras85D*, which is part of the canonical Ras–Raf–MEK signaling cascade downstream of EGFR that leads to the phosphorylation of Erk. We found that FSC clones that were homozygous for *Ras85D$^{[\Delta c40b]}$*, a loss-of-function allele (here referred to as *Ras85D$^-$*) had a phenotype that was similar to, but less penetrant than, *Egfr$^{f24}$* clones (*Figure 7A,E* and *Figure 7—figure supplement 1A–B*). Specifically, we found that 48% (n = 15/31) of *Ras85D$^-$* FSC clones lacked Dlg on the cell surface and had the same morphological defects as *Egfr$^{f24}$* FSC clones, whereas 96% (n = 45/47) of large *Ras85D$^-$* prefollicle cell clones had a normal cell shape and properly localized Dlg to the cell membrane (*Figure 7B,E*).

Next, because LKB1 is also required for cell polarity in the FSC lineage (*Martin and St Johnston, 2003*; *Haack et al., 2013*), we investigated whether EGFR functions upstream of LKB1. First, we found that cell polarity was disrupted in 46% (n = 17/37) of FSC clones that were homozygous for *lkb1$^{[4A4-2]}$*, a deletion allele that is predicted to be a null (*Martin and St Johnston, 2003*) (here referred to as *lkb1$^-$*), whereas polarity was not disrupted in 93% (n = 62/67) of *lkb1$^-$* prefollicle cell clones (*Figure 7C–E* and *Figure 7—figure supplement 1C*). When activated, LKB1 can phosphorylate AMPK at multiple sites, including threonine-184 (T172 in humans), which can be detected with a monoclonal antibody against the human epitope (*Pan and Hardie, 2002*; *Lizcano et al., 2004*). Therefore, we stained for phosphorylated AMPK (pAMPK) and found that it was detectable in follicle cells of 59% (n = 95/160) of wildtype germaria (*Table 2*). Next, we stained for pAMPK in germaria with *lkb1$^-$* FSC clones. We found that, whereas pAMPK signal was absent in 100% (n = 57/57) of *lkb1$^-$* FSC clones, it was clearly detectable in the wildtype (*lkb1$^{+/-}$*) follicle cells in 68% (n = 39/57) of these same germaria (*Figure 7—figure supplement 1D–E*). These data confirm that the LKB1–AMPK pathway is active in follicle cells within the germarium, and that LKB1 promotes the establishment of follicle cell polarity.

LKB1 is activated by PKA phosphorylation, and *lkb1$^{S535E}$* is a constitutively active allele with a serine-535 to glutamic acid substitution that mimics phosphorylation by PKA (*Martin and St Johnston, 2003*). To determine whether *lkb1$^{S535E}$* can rescue the polarity defects caused by loss of EGFR, we investigated the phenotypes in germaria that express either *Egfr$^{DN}$* alone, or both *Egfr$^{DN}$* and *lkb1$^{S535E}$* together under the control of 109-30-Gal4. We found that co-expression of *lkb1$^{S535E}$* with *Egfr$^{DN}$* produced an approximately threefold reduction in the percentage of germaria with follicle cell polarity defects (14%, n = 21/154, *Figure 7F–I*), indicating that constitutive activation of LKB1 partially rescued the *Egfr* mutant polarity defect.

To determine whether EGFR is required for LKB1–AMPK signaling in follicle cells, we stained germaria expressing *Egfr$^{DN}$* in early follicle cells for pAMPK (*Figure 7J*). We observed a 29% decrease in germaria with detectable pAMPK (42%, n = 28/66, *Table 2*) and found a strong correlation between the absence of pAMPK signal and the absence of Dlg on the cell membranes in germaria expressing *Egfr$^{DN}$* (*Figure 7K* and *Table 2*, p < 10$^{-4}$). Collectively, these data indicate that EGFR functions through both the canonical Ras–Raf–MEK–Erk pathway and the LKB1–AMPK pathway to establish epithelial polarity in the FSC.

## Discussion
In this study, we found that EGFR signaling is required in an epithelial stem cell population, but not in its daughter cells, to facilitate the development of apical–basal polarity throughout the lineage.

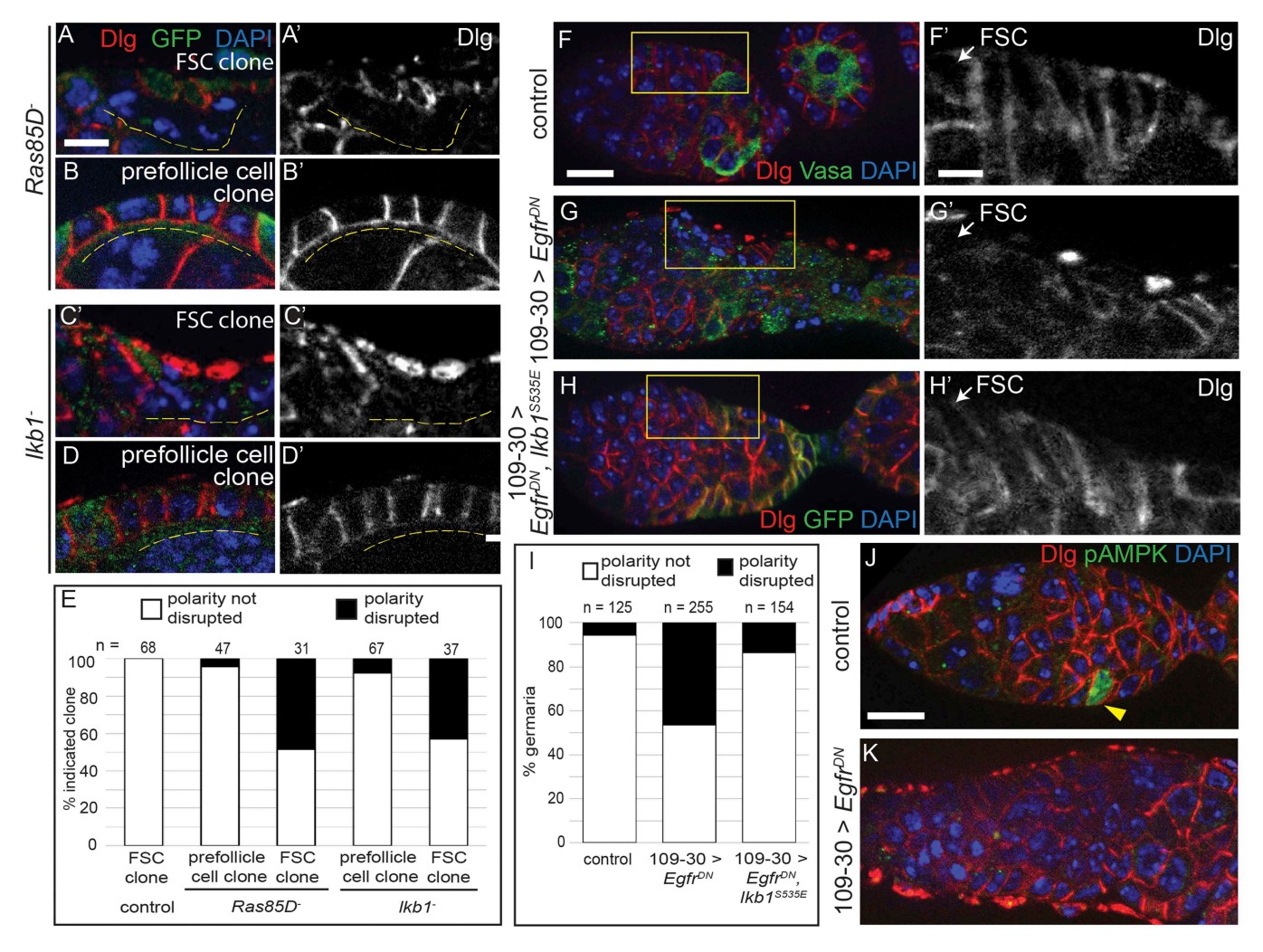

**Figure 7**. EGFR functions upstream of Ras and LKB1 to establish epithelial polarity. (**A–D**) GFP$^{(-)}$ *Ras85D$^-$* (**A**) or *lkb1$^-$* (**C**) FSC clones and *Ras85D$^-$* (**B**) or *lkb1$^-$* (**D**) prefollicle cell clones stained for Dlg (red) and GFP (green). (**E**) Graph indicating the frequencies of polarity phenotypes in wildtype, *Ras85D$^-$*, and *lkb1$^-$* FSC clones and in *Ras85D$^-$* and *lkb1$^-$* prefollicle cell clones. (**F–H**) Germaria containing UAS-*Egfr$^{DN}$*/UAS-*Egfr$^{DN}$* but no Gal4 driver (**F**), 109-30-Gal4 and UAS-*Egfr$^{DN}$*/UAS-*Egfr$^{DN}$* (**G**), or 109-30-Gal4, UAS-*Egfr$^{DN}$*/UAS-*Egfr$^{DN}$* and UAS-GFP-*lkb1$^{S535E}$* (**H**) stained for Dlg (red), and either Vasa (green, **F–G**) or GFP (green, **H**). Dlg localization is disrupted in the germaria overexpressing *Egfr$^{DN}$* only (**G**), but it is restored in germaria overexpressing both *Egfr$^{DN}$* and *lkb1$^{S535E}$* (**H**) (**I**) Graph indicating the frequencies of polarity phenotypes in control, *Egfr$^{DN}$* only, and *Egfr$^{DN}$ and lkb1$^{S535E}$* co-expressing germaria. (**J–K**) Germaria containing UAS-*Egfr$^{DN}$*/UAS-*Egfr$^{DN}$* but no Gal4 driver (**J**) or 109-30-Gal4 and UAS-*Egfr$^{DN}$*/UAS-*Egfr$^{DN}$* (**K**) stained for Dlg (red) and pAMPK (green), which is detectable in prefollicle cells of the control (yellow arrowhead, **J**) but not in germaria overexpressing *Egfr$^{DN}$* (**K**). Images in (**A–D**) are oriented with the apical surface of the follicle cells on the bottom, and GFP$^{(-)}$ clones are indicated by dashed yellow lines. Panels **A'–D'** show the red channel only. Boxed regions of **F–H** are magnified in **F'–H'**, and white arrows indicate the position of the FSC niche. All tissues stained with DAPI (blue). Anterior is to the left in **F–K**. Scale bar represents 5 μm in **F–K** and 1 μm in **A–D** and in magnified insets.

The following figure supplement is available for figure 7:

**Figure supplement 1**. Polarity phenotypes of *Ras85D−* and *lkb1−* FSC clones.

Our finding that EGFR signaling is active specifically in FSCs is supported by our stains for pErk (which we detected specifically in FSCs) and by our mosaic analysis (which indicated that deletion of *Egfr* from FSCs produces a cell-polarity phenotype, whereas deletion of *Egfr* from the immediate daughter cells does not). In our subsequent analysis, we found that loss of *Egfr* from FSCs disrupts the basal and lateral domains of the FSCs (as well as their daughters), whereas constitutive activation of *Egfr* is sufficient to suppress the formation of the apical domains in prefollicle cells. Collectively

**Table 2.** Quantification of the correlation between pAMPK and polarity phenotypes in control or *Egfr^DN*-expressing early follicle cells

|  | Control | | 109–30 > *Egfr^DN* | |
| --- | --- | --- | --- | --- |
|  | pAMPK on | pAMPK off | pAMPK on | pAMPK off |
| Polarity not disrupted | 59%, n = 95/160 | 38%, n = 60/160 | 42%, n = 28/66 | 21%, n = 14/66 |
| Polarity disrupted | 1%, n = 1/160 | 2%, n = 4/160 | 0%, n = 0/66 | 37%, n = 24/66 |
| p-values | 0.0836 | | <0.0001 | |

Values reflect both the percent and fraction of germaria containing either UAS-*Egfr^DN*/UAS-*Egfr^DN* but no Gal4 driver (control), or 109-30-Gal4 and UAS-*Egfr^DN*/UAS-*Egfr^DN*, in which follicle cell polarity is either disrupted or not disrupted (indicated by absence or presence of Dlg on the cell membrane, **Figure 7K**), and in which pAMPK is either detectable (pAMPK on) or absent (pAMPK off). p-values were determined using a two-tailed Fisher's exact test.

these observations support a model (**Figure 8**) in which EGFR signaling promotes basal and lateral identity and suppresses apical identity in FSCs.

It is interesting that, despite the importance of EGFR signaling in FSCs, *Egfr* is dispensable for the maintenance and continued development of cell polarity in prefollicle cells (**Figure 5**). In polarized epithelial cells, apical and lateral identities are maintained by a process of mutual exclusion in which the

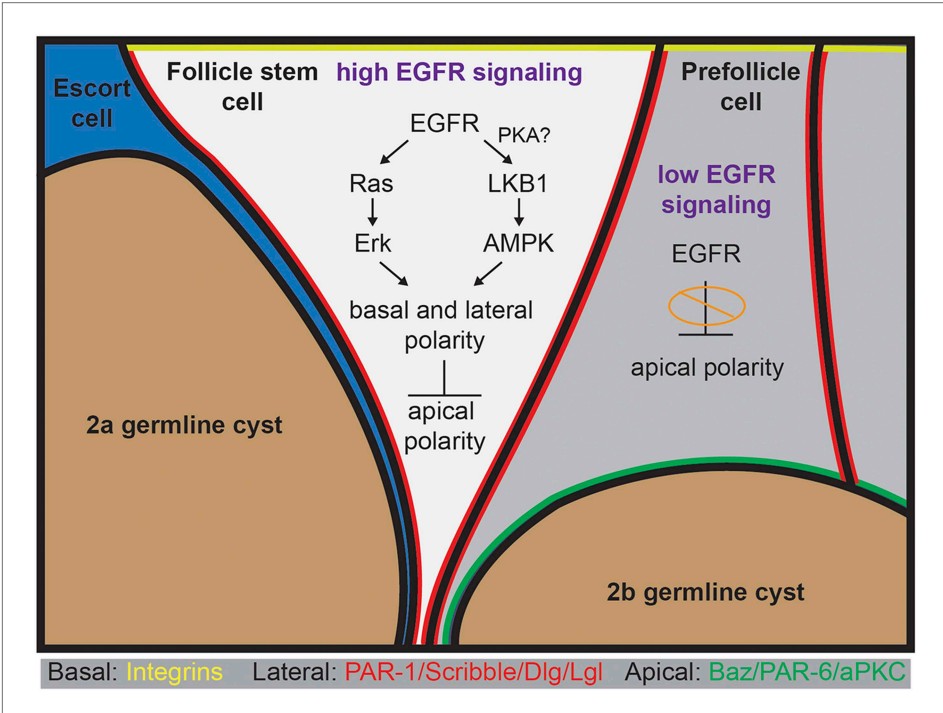

**Figure 8**. A model for the role of EGFR in the establishment of epithelial polarity. High levels of EGFR signaling in the FSC promote maintenance in the niche and the formation of basal and lateral domains while suppressing the formation of an apical domain. EGFR activates both the canonical Ras-mediated pathway leading to the phosphorylation of Erk, and the LKB1–AMPK pathway. Both Erk and AMPK are kinases that can regulate gene activity by activating transcription factors and phosphorylating proteins in the cytoplasm. AMPK directly promotes the lateral identity in polarized cells by activating lateral proteins. PKA is an upstream activator of LKB1 in follicle cells, and PKA can be activated by EGFR signaling, suggesting that EGFR signaling may activate LKB1 via PKA. EGFR signaling may suppress apical polarity either directly by regulating the transcription or activity of apical proteins, or indirectly by enhancing the activity of lateral proteins that suppress the localization of apical proteins. Low levels of EGFR signaling in prefollicle cells relieves this suppression, allowing apical domains to form and permitting differentiation away from the stem cell fate.

cortical localization of the Par-6/aPKC complex is suppressed by Lgl and vice versa (**Hutterer et al., 2004**). Thus, it may be that EGFR signaling is required to maintain the unique partially polarized state of the FSCs but, once both apical and lateral domains are present together, they become self-sustaining. Indeed, the reactivation of EGFR signaling in follicle cells during mid- and late-oogenesis does not seem to disrupt polarity, suggesting that the influence of EGFR signaling on cell polarity is diminished in these stages when the cells are more differentiated and fully polarized. This mutual dependency may also explain why prefollicle cells produced by FSCs that are mutant for *Egfr* exhibit cell-polarity defects (**Figure 3**). The segregation of the basal and lateral protein complexes during an FSC division has not been characterized, but it is plausible that prefollicle cells inherit their basal and lateral domain identities from the FSC. Since *Egfr⁻* FSCs lack basal and lateral domain identities, they may be unable to contribute polarity information to their prefollicle cell daughters, leaving these cells with no polarity cues to build upon and therefore rendering them incapable of specifying basal, lateral or apical domains.

Several lines of evidence indicate that EGFR signaling regulates cell polarity in part by activating LKB1. First, our observations that loss of *lkb1* phenocopies loss of *Egfr*, and that constitutively active LKB1 partially rescues the phenotypes caused by a decrease in *Egfr* function, indicate that EGFR and LKB1 operate together to promote cell polarity in the FSC lineage. Second, our observation that the LKB1-dependent phosphorylation of AMPK is dependent in part on *Egfr* suggests that LKB1 is activated by EGFR signaling. LKB1 is a 'master regulator' of cell polarity (**Partanen et al., 2013**), capable of initiating apical–basal polarity even in cultured cells that lack cell–cell contacts (**Baas et al., 2004**). In *Drosophila*, LKB1 is required in follicle cells to prevent apical proteins from encroaching into the lateral domain (**Martin and St Johnston, 2003**). Moreover, although the intermediate steps were not investigated, a recent study demonstrated that EGFR promotes apical constriction of epithelial cells in the tracheal placode during *Drosophila* development (**Kondo and Hayashi, 2013**).

In many *Drosophila* cell types, EGFR signaling operates exclusively through the linear Ras–Raf–MEK–Erk cascade (**Perrimon et al., 2012**; **Shilo, 2014**). Thus, it is possible that the activation of LKB1 in the FSC lineage is also induced by pErk, which could promote the transcription or post-translational activation of either LKB1 or an upstream activator of LKB1. However, our finding that the loss of *Ras85D* produces a substantially less penetrant phenotype than loss of *Egfr* (**Figure 7E**) suggests that the pathway bifurcates upstream of *Ras85D*, placing *lkb1* in a separate pathway downstream of EGFR. Consistent with this possibility, a study of the adult *Drosophila* brain found that EGFR can activate PKA in a Ras-independent manner (**Hannan et al., 2006**), and PKA is a well-established activator of LKB1 (**Collins et al., 2000**; **Sapkota et al., 2001**). Moreover, each of the individual steps of this pathway from EGFR through to PKA, LKB1, and AMPK to apical–basal polarity have been observed in studies of mammalian tissues (**Collins et al., 2000**; **Tortora and Ciardiello, 2002**; **Xie et al., 2006**; **Shackelford and Shaw, 2009**), suggesting that these connections make up an evolutionarily conserved pathway.

In addition to promoting apical–basal polarity, our findings indicate that EGFR signaling also regulates the segregation of stem cell and daughter cell fates in the FSC lineage. Specifically, our finding that *Egfr^f24^* FSC clones are rapidly lost from the tissue indicates that EGFR is required for the FSC fate; whereas our observation that constitutively active EGFR signaling causes prefollicle cells to retain an FSC-like morphology and replace wildtype stem cells more often suggests that EGFR signaling must be downregulated to permit differentiation. EGFR signaling may promote the FSC fate in several ways. First, the activation of the canonical EGFR pathway leading to the phosphorylation of Erk is likely to directly regulate the activity of many genes in the FSC self-renewal program. Second, EGFR signaling may interact with other pathways, such as the Wingless pathway, that are required for FSC self-renewal (**Song and Xie, 2003**; **Sahai-Hernandez and Nystul, 2013**). Indeed, EGFR and Wingless signaling cooperate to specify cell fate in other *Drosophila* tissues such as the wing disc (**Szüts et al., 1997**) and the intestinal epithelium (**Xu et al., 2011**), and thus there may be similar cross-talk in the FSC niche compartment. Third, our data strongly suggest that EGFR signaling is required for FSC maintenance in the niche in part because of its role in regulating cell polarity. Specifically, our observation that DE-cad and β-integrin are absent from the membranes of cells in *Egfr^f24^* FSC clones indicates that EGFR signaling is required for the formation of the cellular junctions that are known to anchor FSCs in the niche (**Song and Xie, 2002**; **O'Reilly et al., 2008**). In addition, the lack of polarity in *Egfr^f24^* FSCs could also affect other processes, such as cellular trafficking, cell division, and signal transduction that may be important for the self-renewal program.

Collectively these studies demonstrate that EGFR signaling, and the role that the pathway plays in regulating cell polarity, are an essential part of the program that promotes the segregation of FSC and

daughter cell fates. EGFR signaling is known to promote a stem-like or less differentiated state in other epithelial tissues as well. For example, EGFR signaling induces proliferation of progenitor cells in the basal layer of the interfollicular epidermis, and is downregulated in cells within the suprabasal layers that are differentiating into mature keratinocytes (*Wang et al., 2006*). Likewise, EGFR signaling is required for the maintenance and proliferation of the stem cells in the *Drosophila* intestinal epithelium (*Xu et al., 2011*). Moreover, EGFR signaling is commonly upregulated in epithelial cancers, such as triple negative breast cancer, that have a stem cell-like molecular profile. However, the specific effects of EGFR signaling in the stem cells of these tissues are not well understood. In addition, although EGFR signaling can promote a loss of cell polarity in differentiated epithelial cells by contributing to the activation of the epithelial-to-mesenchymal transition, it is unclear whether or how this function of EGFR signaling relates to cell fate specification. Our study suggests that EGFR signaling promotes the stem cell fate at least in part by specifying the unique polarity of the stem cell. It will be interesting to determine whether the regulation of cell polarity is a common mechanism by which cell fates are specified in other epithelial tissues as well.

## Materials and methods

### Fly stocks

Fly stocks were maintained on standard molasses food.

### The following genotypes were used to generate clones

GFP$^{(-)}$ clones: (1) wildtype control for *Egfr$^{f24}$*: FRT 42d/FRT 42d, Ubi-GFP; MKRS(hsFlp)/+, (2) *Egfr$^{f24}$*: FRT 42d, Egfr[f24]/FRT 42d, Ubi-GFP; MKRS(hsFlp)/+, (3) wildtype control for *Ras85D$^-$* and *lkb1$^-$*: hsFlp/+; FRT 82b/FRT 82b, Ubi-GFP, (4) *Ras85D$^-$*: hsFlp/+; FRT 82b, Ras85D[Δc40b]/FRT 82b, Ubi-GFP, (5) *lkb1$^-$*: hsFlp/+; FRT 82b, lkb1[4A4-2]/FRT 82b, Ubi-GFP.

MARCM clones: (1) wildtype control for *Egfr$^{λtop}$*: hsFlp, tub-Gal4, UAS-GFP/+; FRT 40a/tub-Gal80, FRT 40a, (2) *Egfr$^{λtop}$*: hsFlp, tub-Gal4, UAS-GFP/+; FRT 40a/tub-Gal80, FRT 40a; UAS-λtop/+, (3) wildtype control for *Egfr$^{f2}$*: hsFlp, tub-Gal4, UAS-GFP/+; FRT 42d/FRT 42d, tub-Gal80, (4) *Egfr$^{f2}$*: hsFlp, tub-Gal4, UAS-GFP/+; FRT 42d, Egfr$^{f2}$/FRT 42d, tub-Gal80.

### The following genotypes were used in Gal4 experiments

(1) 109-30 > *Egfr$^{λtop}$*: P{GawB}109-30/+; P{UAS-Egfr.λtop}4.4/+, (2) 109-30 > *Egfr$^{DN}$*: P{GawB}109-30/P{w[+mC]=UAS-Egfr.DN.B}29-77-1; P{w[+mC]=UAS-Egfr.DN.B}29-8-1/+, (3) 109-30 > *Egfr$^{DN}$*, *lkb1$^{S535E}$*: P{GawB}109-30/P{w[+mC]=UAS-Egfr.DN.B}29-77-1; P{w[+mC]=UAS-Egfr.DN.B}29-8-1/ P{UASp-GFP-lkb1.S535E}.

Wildtype stock used in *Figure 1—figure supplement 1A* was *y[1] w[1]*.

All stocks were obtained from the Bloomington Stock Center except the following: *yw, hsFlp, tub-Gal4, UAS-GFP/FM7; tub-Gal80 FRT40A/CyO* obtained from Yuh Nung Jan, *w; FRT 42D, Ubi-GFP/ CyO; MKRS(hsFlp)/TM2* obtained from Allan Spradling, *FRT 42d, EGFR[f24]/CyO* and *FRT 82b, Ras85D[ΔC40b]/TM3* obtained from Bruce Edgar, *hsFlp, tub-Gal4, UAS-GFP; FRT 42d, tub-Gal80* obtained from Ben Ohlstein, *w; P{UASp-GFP-lkb1.S535E}TM6B* and *w; FRT 82b, lkb1[4A4-2]/TM3* obtained from Daniel St Johnston, and *P{UAS-Egfr.λtop}4.4* obtained from Trudi Schupbach.

### Clone induction experiments

Clones were generated by culturing flies of the appropriate genotypes and carrying control and experimental adults as paired cohorts through an identical clone induction process as follows: adults were given wet yeast for 2 days at 25°C, heat shocked twice a day for 2 days (4 times total) for 1 hr in a 37°C water bath, then maintained on wet yeast at 25°C for up to 21 days post heat shock, and dissected at the indicated days post clone induction. Wet yeast was changed daily.

### FSC vs transient clones

We took advantage of the following characteristics to differentiate between FSC and transient clones. When a clone is induced in an FSC, the labeled FSC remains in the niche and continues to divide as the clone grows (*Margolis and Spradling, 1995*). Therefore, these clones can span across many follicles and will always include at least one cell at the Region 2a/2b border, where the FSC niche is located. In contrast, when a clone is induced downstream from the FSC (for example in a prefollicle cell produced by an FSC), the clone will move out of the germarium as it grows; it will not span across

more than two follicles and will not cover more than approximately one half of a single follicle (*Nystul and Spradling, 2010*). Occasionally, the FSC that is maintaining an FSC clone is lost from the niche due to stem cell replacement, which results in a (former) stem cell clone that no longer extends back to the Region 2a/2b border. These can be difficult to distinguish from FSC clones unless they are very big (e.g. clearly spanning three or more consecutive follicles). However, *Egfr⁻* clones that originate from an FSC should have a polarity phenotype, yet only 5–6% of *Egfr⁻* prefollicle cell clones scored at 4–7 dpci had polarity defects (*Table 1* and *Figure 3—figure supplement 2*), indicating that this type of 'false' transient clone is uncommon at these time points.

### Gal4 experiments

Experimental flies bearing the 109-30-Gal4 construct were raised at 25°C and then shifted to 29°C 1–2 days post eclosion and maintained on wet yeast. They were dissected at 18–21 days post eclosion.

Where p values are indicated, phenotypes were quantified and significance was determined with a two-tailed t-test, or a two-tailed Fisher's exact test.

### Immunostaining

To ensure that oogenesis proceeded regularly and that we minimized any potentially complicating influences of starvation, all adult flies used were given fresh wet yeast daily. For pErk staining, adult flies of the appropriate genotype were given fresh wet yeast 4–6 hr before dissecting. Ovaries were dissected in Graces medium (Gemini Bio-Products, West Sacramento, CA) using tungsten needles and carefully transferred to microfuge tubes with forceps or a tungsten needle. To minimize mechanical damage, ovarioles were never aspirated with a pipette. Ovarioles were fixed in 1× PBS + 4% paraform-aldehyde diluted from 16% (Fisher) for 15 min, rinsed, and incubated with primary antibodies overnight at 4°C. Tissues were then rinsed and washed for 1 hr, incubated with secondary antibody at room temperature for 2 hr, rinsed and washed for 1 hr, rinsed in 1× PBS, and mounted in Vectashield plus DAPI (Vector Labs) on glass slides. 1× PBST (PBS + 0.2% Triton X-100) was used for all antibody dilutions, rinses and washes except where indicated otherwise. All images were acquired on a Zeiss M2 Axioimager with Apotome unit or Leica TCS SP5 spectral confocal. For multicolor fluorescence images, each channel was acquired separately. Images were stored as JPEG files and post-acquisition processing, such as adjustments of brightness, rotations and cropping, was performed with Adobe Photoshop.

The following primary antibodies were used: From Cell Signaling (Danvers, MA): rabbit anti-phosphorylated Erk1/2 (Thr202/Tyr204) (4370, 1:200), rabbit anti-phosphorylated AMPKα (2535, 1:200), and αvδ rabbit anti-cleaved caspase 3 (9661S, 1:200). From Developmental Studies Hybridoma Bank (Iowa City, Iowa): mouse anti-Dlg (4F3, 1:200), rat anti-DE-cadherin (DCAD2, 1:100), mouse anti-integrin βPS (CF.6G11, 1:100) and mouse anti-FasIII (7G10, 1:50). From Santa Cruz Biotechnologies (Santa Cruz, CA): anti-rabbit aPKC (SC-216, 1:50) and anti-rabbit Vasa (SC-30210, 1:1000). rabbit anti-GFP (Torrey Pines Biolabs, Secaucus, New Jersey, TP401, 1:5000), mouse anti-GFP (Invitrogen, Grand Island, NY, A11120, 1:100), rabbit anti-Bazooka (1:1000) (a gift from Andreas Wodarz), and guinea pig anti-traffic jam (1:4000) (a gift from Allan Spradling). The following secondary antibodies were used: anti-rabbit and anti-mouse conjugated to Alexa Fluor 488, 546, or 555 (Invitrogen A11001, A11008, A11010, or A21424, 1:1000).

## Acknowledgements

We are grateful to Marco Conti, Ophir Klein, and Matt Cook for critical comments on the manuscript and to our colleagues in the fly community (cited in the methods section) for fly stocks and antibodies. We also thank the Bloomington Stock Center and the Developmental Studies Hybridoma Bank for curation of many stocks and reagents used in this study. This work was supported by NIH Grant R01 GM097158 to T.G.N.

## Additional information

### Funding

| Funder | Grant reference number | Author |
| --- | --- | --- |
| National Institute of General Medical Sciences | R01GM097158 | Angela Castanieto, Michael J Johnston, Todd G Nystul |

| Funder | Grant reference number | Author |
|---|---|---|
| National Institute of General Medical Sciences | T32GM007810 | Angela Castanieto, Michael J Johnston |
| University of California | Genentech Predoctoral Fellowship | Angela Castanieto, Michael J Johnston |
| University of California, Los Angeles | Eugene Cota Robles Fellowship | Angela Castanieto |

The funders had no role in study design, data collection and interpretation, or the decision to submit the work for publication.

### Author contributions

AC, MJJ, TGN, Conception and design, Acquisition of data, Analysis and interpretation of data, Drafting or revising the article

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
