## [Decision Letter]

Thank you for sending your work entitled “EGFR signaling promotes self-renewal through the establishment of cell polarity in *Drosophila* follicle stem cells” for consideration at *eLife*. Your article has been favorably evaluated by Fiona Watt (Senior editor), a Reviewing editor, and 3 reviewers.

The Reviewing editor and the reviewers discussed their comments before we reached this decision, and the Reviewing editor has assembled the following comments to help you prepare a revised submission.

The general consensus is that this paper is in principle appropriate for *eLife* as it is novel and addresses issues of substantial general interest. There are, however, substantial concerns about whether the main conclusions are rigorously supported that would need to be addressed before it could be accepted.

The main experimental approach is to generate follicle cell clones and this is where several issues arise. First, there is the possibility that some of the clones may be 'false' as recently described by the St Johnston lab. Second, it is not clear how they distinguish between stem cell clones and prefollicle cell clones in Figures 3 and 4, given that many of the mutant stem cells are lost, so one cannot always identify the former by the presence of a GFP-negative FSC. Third, only a single allele of each mutation has been tested and there is a risk of contaminating mutations. This is somewhat abrogated by the use of lambda-top, but here, too, only a single insertion was tested. The DN-EGFR, Ras85D and LKB1 effects were rather weak.

All these issues could be addressed by generating one or two positively marked MARCM clones, using another allele. Were the results to be clear, it might not be necessary to do it for all the genes tested.

The exact role of EGFR is not easy to conclude from these experiments. It appears to have distinct functions in FSCs and PFCs. The authors should be more circumspect; it is probably premature to describe everything in terms of a simple linear pathway through LKB1. Furthermore, the strong statement that “The EGFR is required for FSC self-renewal” is misleading, as half of the *Egfr–* FSC clones persist for 11 days. During this time, they must go through multiple rounds of asymmetric division to produce a daughter stem cell and a prefollicle cell, that is, self-renew. What the authors are referring to is the observation that mutant stem cells cannot replace their counterparts on the other side of the germarium, which is stem cell replacement, not self-renewal.

---

## [Author Response]

*The main experimental approach is to generate follicle cell clones and this is where several issues arise. First, there is the possibility that some of the clones may be 'false' as recently described by the St Johnston lab. Second, it is not clear how they distinguish between stem cell clones and prefollicle cell clones in*
Figures 3 and 4*, given that many of the mutant stem cells are lost, so one cannot always identify the former by the presence of a GFP-negative FSC*.

We have added a section to the Methods section titled “FSC versus transient clones” that provides additional clarification of this point.

*Third, only a single allele of each mutation has been tested and there is a risk of contaminating mutations. This is somewhat abrogated by the use of lambda-top, but here, too, only a single insertion was tested. The DN-EGFR, Ras85D and LKB1 effects were rather weak*.

*All these issues could be addressed by generating one or two positively marked MARCM clones, using another allele. Were the results to be clear, it might not be necessary to do it for all the genes tested*.

As suggested, we generated positively marked MARCM clones using another null allele of *Egfr* (allele F2) and assayed for cell-polarity defects. This experiment provided a clear confirmation of our previous results: as with the F24 allele, nearly all F2 FSC clones (94%, n = 15/16) had defective polarity whereas polarity was not substantially disrupted in nearly all of the F2 transient clones (95%, n = 112/118). These results are presented in Figure 3—figure supplement 1 and 2.

*The exact role of EGFR is not easy to conclude from these experiments. It appears to have distinct functions in FSCs and PFCs*.

Our data support a model in which EGFR functions in FSCs to maintain basal and lateral polarity of FSCs but is not required in prefollicle cells for the maintenance or continued development of cell polarity. However, we also found that prefollicle cells produced by *Egfr*^*–*^ FSCs have polarity defects even though these cells do not need EGFR to maintain polarity. This may be because prefollicle cells produced by *Egfr*^*–*^ FSCs do not inherit the proper polarity cues and thus cannot become polarized. We have added a discussion of this issue in the Discussion section, which includes the following possible explanation:

“The segregation of the basal and lateral protein complexes during an FSC division has not been characterized, but it is plausible that prefollicle cells inherit their basal and lateral domain identities from the FSC. Since *Egfr*^*–*^ FSCs lack basal and lateral domain identities, they may be unable to contribute polarity information to their prefollicle cell daughters, leaving these cells with no polarity cues to build upon and therefore rendering them incapable of specifying basal, lateral or apical domains.”

*The authors should be more circumspect; it is probably premature to describe everything in terms of a simple linear pathway through LKB1*.

Our favored model (Figure 8) is not a linear pathway through LKB1 but instead that LKB1 is a member of just one of the branches downstream from EGFR in FSCs. In addition, we have tried to be very cautious throughout the manuscript to leave open the possibility that other proteins may function downstream of EGFR to regulate cell polarity. For example, we state in the Discussion section “Several lines of evidence indicate that EGFR signaling regulates cell polarity *in part* by activating LKB1” and go on to point out that “constitutively active LKB1 *partially* rescues the phenotypes caused by a decrease in *Egfr* function”. This discussion continues in the following paragraph, where we describe the bifurcation model using language that emphasizes our uncertainty, such as “it is possible that,” “suggests,” and “consistent with this possibility”. Also, in the next paragraph of the same section, we discuss the many ways that the phosphorylation of Erk may directly regulate the activity of genes in the FSC self-renewal program. We hope this is an effective way of communicating our views without overstating them.

*Furthermore, the strong statement that “The EGFR is required for FSC self-renewal” is misleading, as half of the Egfr*^*–*^
*FSC clones persist for 11 days. During this time, they must go through multiple rounds of asymmetric division to produce a daughter stem cell and a prefollicle cell, that is, self-renew. What the authors are referring to is the observation that mutant stem cells cannot replace their counterparts on the other side of the germarium, which is stem cell replacement, not self-renewal*.

We consider the term “self-renewal” to encompass not just the ability of a stem cell to divide asymmetrically but also the ability of the stem cell to remain in the niche and continue dividing in this way over a period of time that is comparable to that of a wildtype stem cell. Thus, we interpret our observation that *Egfr*^*f24*^ FSC clones are lost from the tissue much more rapidly than wildtype FSC clones as evidence that *Egfr* is required for FSC self-renewal. However, we understand from the reviewers’ comment that this wording can be ambiguous, so we have removed “self-renewal” from the title and replaced most instances of this term with “maintenance in the niche” or “stem cell replacement”, as the reviewers suggest.